# The Impact of Multidrug-Resistant *Acinetobacter baumannii* Infection in Critically Ill Patients with or without COVID-19 Infection

**DOI:** 10.3390/healthcare11040487

**Published:** 2023-02-08

**Authors:** Thamer A. Alenazi, Maryam S. Bin Shaman, Durria M. Suliman, Turkiah A. Alanazi, Shoroq M. Altawalbeh, Hanan Alshareef, Doha I. Lahreche, Sayer Al-Azzam, Mohammad Araydah, Reema Karasneh, Faycal Rebahi, Marwah H. Alharbi, Mamoon A. Aldeyab

**Affiliations:** 1Department of Infectious Diseases, King Fahad Specialist Hospital, Ministry of Health, Tabuk 47717, Saudi Arabia; 2Department of Pharmacy, Prince Mohammad Medical City, Ministry of Health, Aljouf 85846, Saudi Arabia; 3Department of Pharmaceutical Care, King Fahd Specialist Hospital, Ministry of Health, Tabuk 47717, Saudi Arabia; 4Department of Clinical Pharmacy, Jordan University of Science and Technology, P.O. Box 566, Irbid 22110, Jordan; 5Department of Pharmacy Practice, Faculty of Pharmacy, University of Tabuk, Tabuk 47717, Saudi Arabia; 6Department of Basic Medical Sciences, Faculty of Medicine, Yarmouk University, P.O. Box 566, Irbid 22110, Jordan; 7Department of Intensive Care Unit, King Fahad Specialist Hospital, Ministry of Health, Tabuk 47717, Saudi Arabia; 8Radiology Department, King Fahad Specialist Hospital, Ministry of Health, Tabuk 47717, Saudi Arabia; 9Department of Pharmacy, School of Applied Sciences, University of Huddersfield, Huddersfield HD1 3DH, UK

**Keywords:** multidrug-resistant *Acinetobacter baumannii*, critically ill patients, intensive care unit, COVID-19 infection

## Abstract

This is a single-center, retrospective, cohort study aimed to evaluate the clinical outcomes of multi-drug resistance in *Acinetobacter baumannii* infections (MDR-AB) in intensive care unit (ICU) patients with or without a COVID-19 infection and risk factors for blood stream infection. A total of 170 patients with MDR-AB were enrolled in the study. Of these, 118 (70%) patients were admitted to the ICU due to a COVID-19 infection. Comparing the COVID-19 and non-COVID-19 groups, the use of mechanical ventilation (98.31% vs. 76.92%, *p* = 0.000), the presence of septic shock (96.61% vs. 82.69%, *p* = 0.002), and the use of steroid (99.15% vs. 71.15%, *p* = 0.000) and tocilizumab therapies (33.05% vs. 0%, *p* = 0.000) were more prevalent and statistically more significant in patients with COVID-19 infections. The average length of the ICU stay (21.2 vs. 28.33, *p* = 0.0042) was significantly lower in patients with COVID-19 infections. Survival rate was 21.19% for the COVID-19 group and 28.85% for non-COVID-19 group with a *p*-value = 0.0361. COVID-19 status was associated with significantly higher hazards of death (HR 1.79, CI 95% 1.02–3.15, *p* = 0.043). Higher SOFAB (15.07 vs. 12.07, *p* = 0.0032) and the placement of an intravascular device (97.06% vs. 89.71%, *p* = 0.046) were significantly associated with the development of a bloodstream infection. Our study has shown that critically ill patients with an MDR-AB infection, who were admitted due to a COVID-19 infection, had a higher hazard for death compared to non-COVID-19 infected patients.

## 1. Introduction

*Acinetobacter baumannii* is an aerobic, Gram-negative opportunistic, glucose non-fermentative and non-motile coccobacillus commonly found in various environments, such as soil and water [1]. This bacterium can adhere to medical devices (including the system used for mechanical ventilation) and survive up to 33 days on dry surfaces [2]. Two percent of hospital-associated infections in the United States (US) and Europe is caused by AB infection [3,4]. In the Middle East and Asia, the incidence of this infection is two times higher [4]. Approximately 45% of A. baumannii infections is multidrug-resistant (MDR), which is four times higher than the rate of developing MDR in other Gram-negative bacteria [5]. The acquisition of multiple drug resistance, especially to carbapenems, has made this pathogen a major public health concern [6]. The negative impact on outcomes and high mortality rates associated with this infection is due to inappropriate therapies and limited therapeutic options, especially in ICU patients [7,8].

At the end of December 2019, several cases of acute respiratory syndrome were reported in Wuhan City, China. A novel coronavirus, called severe acute respiratory syndrome coronavirus 2 (SARS-CoV-2), has been identified as the main causative agent. The disease is now referred to as coronavirus disease 2019 (COVID-19) [9,10]. The clinical characteristics of COVID-19 are manifested in several symptoms ranging from asymptomatic infection to severe illness with a high risk of mortality [11]. COVID-19 can severely damage the lung epithelium and indirectly activate an aberrant “cytokine storm”, leading to multi-organ failure [12,13]. To avoid this activation of the immune system, immunosuppressive drugs are widely used [13,14]. Treatment of COVID-19 patients with cytokine-targeting drugs might be associated with an increased probability of developing MDR infections [15]. The prevalence of co-infection was variable among COVID-19 patients in different studies; however, it could be up to 50% among non-survivors [16].

Superinfection with multidrug-resistant *Acinetobacter baumannii* in ICU COVID-19 patients has been addressed by several studies. A study from the French ICU has demonstrated that 28% of severe SARS-CoV2 pneumonia patients have bacterial coinfections with *Acinetobacter baumannii* [17]. One study conducted by Duployez et al. (2020) showed that 16% of all admitted COVID-19 patients presented with fungal or bacterial coinfection, and *Acinetobacter baumannii* was the most common causative pathogen in respiratory infections and bacteremia [18]. In addition, a study from China has reported that *Acinetobacter baumannii* and *S. aureus* were more frequent pathogens of respiratory coinfections in COVID-19 patients during ICU admission [19]. One study has compared coinfection among ICU patients with or without COVID-19 and found that a carbapenem-resistant *Acinetobacter baumannii* was a more frequent isolate bacterium in COVID-19 patients compared to non-COVID-19 patients [20]. An Italian study has highlighted the risk factors of *Acinetobacter baumannii* acquisition [21], finding that steroid use in COVID-19 patients and previous colonization of this organism could be risk factors for developing multidrug-resistant *Acinetobacter baumannii* infections [21].

As coinfection with AB in COVID-19 patients has been reported several times in the literature, and as infections with this bacterium significantly increase the morbidity and mortality, this study aimed to evaluate the clinical outcomes of MDR *Acinetobacter* infection in critically ill patients with or without a COVID-19 infection. In addition, to determining the factors that are associated with increased mortality and a bloodstream infection in this group of patients. This will help clinicians implement early and correct interventions that could reduce mortality.

## 2. Methods

### Study Design and Patient Selection

This is a single-center, retrospective cohort study that included patients with an MDR-AB infection admitted to the King Fahad Specialist Hospital in Tabuk in Saudi Arabia, for the period between March 2020 and September 2021. This center is the largest referral hospital in the Tabuk region, with a 500-bed capacity. Patients were divided into two groups: patients diagnosed with and without COVID-19 who were admitted to the ICU. Inclusion criteria for all patients are (1) age ≥ 18 years; (2) ICU admission during the study period; and (3) positive culture of MDR-AB. The COVID-19 group patients were primarily admitted to the ICU because of their COVID-19 status.

The data were collected from the hospital’s computer system and from medical charts. The following information was determined: demographic information; diagnoses during hospital stay; medications used to treat COVID-19; antimicrobials used to treat the superimposing infection; radiological imaging in the form of chest X-ray findings in three sequences (upon admission, upon positive culture and upon discharge); microbiological data; duration of ICU and hospital stay; any infection during hospitalization; use of steroid therapy; sequential organ failure assessment score (SOFA); COVID-19 stage status and procedures, including invasive mechanical ventilation (MV), continuous renal replacement therapy (CRRT) and extracorporeal membrane oxygenation (ECMO); comorbidities [chronic obstructed pulmonary disease (COPD), chronic heart failure (CHF), hypertension (HTN), diabetes mellites (DM), end-stage renal disease (ESRD), chronic kidney disease (CKD), chronic liver disease (CLD), coronary artery disease (CAD), asthma, smoking, body mass index (BMI)] and triage vitals (fever above 37.5, hypoxia, hypotension and tachycardia).

Initial laboratory tests were conducted, and upon positive culture, the following were taken: white blood cell counts (WBCs), lactate, absolute neutrophil count (ANC), absolute lymphocyte count (ALC), neutrophilic lymphocytic ratio (NLR), platelets count (PLT), hemoglobin (HB), C- reactive protein (CRP), D-dimer, ferritin, lactate dehydrogenase (LDH), troponin, albumin and creatinine. All data were collected during hospitalization.

A bacterial infection with MDR-AB was defined as those patients with positive culture from blood, respiratory, or urine specimens correlated with patient clinical condition during specimen collection. The definition of the multi-drug resistance of AB that we followed in our hospital was any strain that was non-susceptible to ≥1 agent in ≥3 antimicrobial categories [22]. The cut-off values for abnormal WBC are when the total WBC was above 11 × 10^3^/µL and for CRP when the result was above 6 mg/L.

Clinical outcomes that were assessed in the current study include inpatient mortality and the development of bloodstream infections. The study protocol was approved by the institutional review board (IRB) of the Tabuk Institutional Review Board (approval File Number: TU-077/0211114). The ethics committee of the Tabuk Institutional Review Board waived informed consent because this was a retrospective study. Patient data were anonymously analyzed to preserve patient privacy.

## 3. Data Analyses

Patients’ baseline characteristics were described using means with standard deviations for continuous data, and frequency with percentages for categorical data. Patients’ demographics and clinical data were compared between COVID-19 and non-COVID-19 patients using the independent sample t-test or Chi-square test, as appropriate. The Kaplan–Meier estimator was applied to evaluate inpatient survival according to COVID-19 status. For the test of equality of the survivor functions, the log-rank test was used. The Cox hazard regression model was conducted to assess potential factors associated with the hazards of inpatient death. Logistic regression was used to assess potential predictors of the development of bloodstream infections. Variables included in both regression models were selected using a backward stepwise process with *p* < 0.2 to stay. All data analyses were conducted using Stata version 17 software (StataCorp. 2021. Stata: Release 17. Statistical Software. College Station, TX, USA: StataCorp LLC.). The statistical significance was set at a two-sided *p* < 0.05.

## 4. Results

A total of 170 patients were admitted to the ICU with an MDR-AB infection and were retrospectively enrolled in the study. The mean age of the total study population was 62.74 years old, and 34.71% of them were females.

### 4.1. MDR-AB and COVID-19 Infections

Of the study population, 118 patients were admitted to the ICU due to a COVID-19 infection with an MDR-AB superinfection, and the remaining 52 patients were admitted for other reasons.

The clinical characteristics of patients with a documented MDR-AB infection with or without a coexistent COVID-19 infection are illustrated in Table 1. Comparing COVID-19 and non-COVID-19 patients, the average BMI (29.67 vs. 27.03, *p* = 0.0429), placement of intravascular device (97.46% vs. 86.54%, *p* = 0.005), use of mechanical ventilation (98.31% vs. 76.92%, *p* ≤ 0.0001), presence of septic shock (96.61% vs. 82.69%, *p* = 0.002) and use of steroid (99.15% vs. 71.15%, *p* ≤0.0001) and tocilizumab (33.05% vs. 0%, *p* ≤ 0.0001) therapies were more prevalent and statistically significant in patients with COVID-19 infections.

More than two comorbidities (19.49% vs. 44.23%, *p* = 0.001), COPD (0.85% vs. 30.77%, *p* = 0.001) and average SOFA at time of admission (4.63 vs. 6.13, *p* = 0.0258) were higher and statistically more significant in patients without a COVID-19 infection. Moreover, the average length of the ICU stay (21.2 vs. 28.33, *p* = 0.0042) was significantly lower in patients with a COVID-19 infection, and this might be explained by the higher overall mortality in those patients. Average days to death from ICU admission for COVID-19 patients was 20.9 (SD = 10.8) days, and 27.7 (SD = 19.4) days for non-COVID-19 patients. No statistically significant difference was observed between the two groups regarding the mean age, gender, pregnancy status, SOFA at positive culture time, CRRT, ECMO, bloodstream infection, laboratory findings, average days to transfer to ICU, length of hospital stay, mortality at 30 days and overall mortality. Overall inpatient mortality was not statistically different according to COVID-19 status; the survival rates were 21.19% for the COVID-19 group and 28.85% for non-COVID-19 group with a *p*-value = 0.489.

The Kaplan–Meier curve for inpatient survival of overall patients infected with an MDR-AB infection with or without a coexistent COVID-19 infection is shown in Figure 1. A total of 130 patients died during hospital admission, with a survival rate of 23.53%. The conducted survival model was based on the time from ICU admission to death status at hospital discharge. COVID-19 and non-COVID-19 patients differed significantly in survival; *p*-value = 0.036.

### 4.2. Mortality Hazards in MDR-AB Patients

COVID-19 status was associated with significantly higher hazards of death, adjusting for potential confounders (hazard ratio (HR) 1.79, CI 95% 1.02–3.15, *p* = 0.043). A higher SOFAB score (HR 1.12, CI 95% 1.07–1.16, *p* ≤ 0.0001), an abnormal WBC count (HR 1.57, CI 95% 1.04–2.37, *p* = 0.033) and CKD (HR 2.01, CI 95% 1.26–3.19, *p* = 0.003) were all associated with significantly higher hazards of death, as shown in Table 2.

### 4.3. MDR-AB and Bloodstream Infection

A comparison between the clinical characteristics of patients with a documented MDR-AB infection with or without a bloodstream infection is illustrated in Table 3. Higher SOFAB (15.07 vs. 12.07, *p* = 0.0032) and placement of an intravascular device (97.06% vs. 89.71%, *p* = 0.046) were significantly associated with the development of a bloodstream infection.

On the other hand, the presence of more than two comorbidities (21.57% vs. 35.29%, *p* = 0.048) and hypertension (59.80% vs. 75.00%, *p* = 0.041) were highly prevalent and statistically significant in patients without a bloodstream infection.

Each one-unit increase in SOFA score at positive culture time was associated with significantly higher odds of developing a bloodstream infection (odds ratio 1.13, CI 95% 1.06–1.21, *p* ≤ 0.001). Other factors such as the presence of a COVID-19 infection, abnormal CRP, and placement of an intravascular device were not significantly associated with developing a bloodstream infection; see Table 4.

## 5. Discussion

In this retrospective cohort study of critically ill patients with an MDR-AB infection, patients who were admitted due to a COVID-19 infection had higher hazard of mortality compared to non-COVID-19 infected patients. We studied the outcomes of acquiring an MDR-AB infection in critically ill patients admitted due to a COVID-19 infection; AB is a common pathogen in the hospital environment of high concern, especially in intensive care units, causing critical conditions that mainly affect the respiratory tract and cause severe infections such as bloodstream infections and ventilator-associated pneumonia [23,24,25,26]. In addition, evidence of increased mortality in patients with COVID-19 and bacterial co-infections was reported [27,28,29]. In our study, 118 (69.4%) patients were admitted due to a COVID-19 infection, with 78.8% mortality. This is consistent with a recently published study, conducted at a tertiary care hospital in Italy, for patients admitted to ICUs who acquired MDR-AB, of which the mortality rate for the COVID-19 infected patients was 81% [21].

In the current study, 170 critically ill patients infected with MDR-AB were included. Male patients were dominant in both groups (69.23 and 63.56%), and this was similar to other published studies [21,30,31]. Obesity was significantly higher in COVID-19 infected patients; this was supported by a systematic review and meta-analysis that included nine studies [32]. The authors concluded that obesity may aggravate COVID-19, and obese patients with COVID-19 were more severely affected and had a worse outcome than those without [32]. In our study, 98.31% of co-infected patients received mechanical ventilation, 96.61% had septic shock, and 97.46% required intravascular devices. These findings highlight the severity of infection in these populations compared to non-COVID-19 infected patients. These results are consistent with previously reported characteristics and outcomes in severe and critically ill patients with a COVID-19 infection [9,31,33]. Currently, steroid therapy has shown some promising results in the management of a severe COVID-19 infection. In RECOVERY trial, the dexamethasone use has lower the mortality by one-third in ventilated patients and by one-fifth in oxygen-treated patients [34]. Treatment with tocilizumab is commonly used in critically ill COVID-19 patients, despite the controversies in efficacy and safety [35,36,37]. National guidelines recommend starting systemic corticosteroids in the management of critically ill COVID-19 infected patients and starting tocilizumab in patients who meet specific criteria [38]. In our study, 99.15% of COVID-19 infected patients received steroids such as dexamethasone. However, MDR-AB infected patients who received steroid therapy had a non-significantly lower risk of mortality (HR 0.5 CI 95% 0.21–1.2 *p* = 0.119). Tocilizumab was used in 33.05% of co-infected patients. Recently published guidance for the management of resistant pathogens suggested treatment with at least two medications for mild to moderate MDR-AB infection, even if the monotherapy agent is active, and then de-escalating to a single agent after achieving clinical improvement [39]. This suggestion was based on limited data supporting monotherapy [39].

The length of the ICU stay was significantly higher in MDR-AB patients with non-COVID-19 infection compared to COVID-19 infected patients (28.33 ± 19.63 vs. 21.2 ± 11.98; *p* = 0.0042). This observation could be explained by the higher mortality rate in the second group (78.81%) which shortened the length of stay. In contrast to the A. Russo et al. study, which showed no difference in length of ICU stay between the two groups’ mean days (22.23 ± 9.53 vs. 22.22 ± 9.65; *p* = 0.610) [21], in our study, the hazard of death in MDR-AB patients admitted with a COVID-19 infection was significantly higher (HR 1.79, CI 95% 1.02–3.15, *p* = 0.043) compared to patients admitted for non-COVID-19 infection reasons. This emphasizes that bacterial coinfection, specifically AB, in critically ill patients may worsen the outcomes in COVID-19 patients [21,30].

In addition, our results showed that MDR-AB infected patients with higher SOFA at positive culture time, an abnormal WBC count and CKD had a higher risk of death. Our study showed a COVID-19 infection is not a factor in acquiring a bloodstream infection with MDR-AB (OR 2.06 CI 95% 0.9–4.72, *p* = 0.086). These findings were in contrast to other published research, which showed a severe COVID-19 infection is a risk factor for acquiring a MDR-AB bloodstream infection (OR 15.1, CI 95% 3.7–40.1, *p* = <0.001) [21]. However, in our study, a high SOFA score at the time of positive culture was a risk for acquiring a bloodstream infection (OR 1.13 CI 95% 1.06–1.21, *p* = 0.000).

This study had some limitations. Firstly, it was retrospective in design, which did not allow us to ascertain a complete dataset; for example, there were missing data for some patients regarding comorbidities. Secondly, this study was conducted in a single center; it would have benefited if it had been conducted in multiple centers. Nevertheless, this center is the largest referral hospital in the Tabuk region.

## 6. Conclusions

Our study has shown that critically ill patients with an MDR-AB infection, who were admitted due to a COVID-19 infection, had a higher hazard for death compared to non-COVID-19 infected patients.

## Figures and Tables

**Figure 1 healthcare-11-00487-f001:**
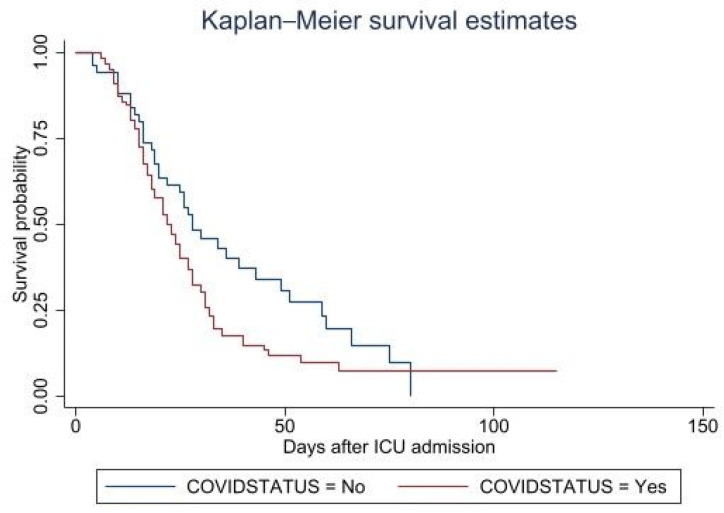
The Kaplan–Meier survival curves from ICU admission to hospital discharge for patients with an MDR-AB infection with or without a COVID-19 infection.

**Table 1 healthcare-11-00487-t001:** The clinical characteristics of patients with a documented MDR-AB infection with or without a coexistent COVID-19 infection.

Variables	Non-COVID-19 *n* = 52	COVID-19 *n* = 118	*p* Value
Age	66.77 ± 20.96	60.97 ± 16.32	0.0525
Gender			0.474
Female (F)	16 (30.77%)	43 (36.44%)	
Male (M)	36 (69.23%)	75 (63.56)	
Body mass index (*n* = 160)	27.03 ± 7.34	29.67 ± 7.18	0.0429
Body mass index category			0.025
Underweight or normal	16 (38.10%)	21 (17.80%)	
Overweight	13 (30.95%)	54 (45.76%)	
Obese	13 (30.95%)	43 (36.44%)	
Pregnancy	0 (0%)	5 (4.24%)	0.132
Comorbidities			
More than two comorbidities	23 (44.23%)	23 (19.49%)	0.001
Heart failure	6 (11.54%)	6 (5.08%)	0.130
Hypertension	35 (67.31%)	77 (65.25%)	0.795
Diabetes Mellitus (DM)	34 (65.38%)	82 (69.49%)	0.596
Chronic kidney disease (CKD)	12 (23.08%)	25 (21.19%)	0.783
Chronic liver disease (CLD)	4 (7.69%)	3 (2.54%)	0.119
Asthma	5 (7.69%)	4 (3.39%)	0.222
Chronic obstructive pulmonary disease (COPD)	16 (30.77%)	1 (0.85%)	0.0001
Malignancy	4 (7.69%)	3 (2.54%)	0.119
Clinical features			
SOFA at time of admission	6.13 ± 3.69	4.63 ± 4.16	0.0258
SOFA at positive culture time	14.23 ± 7.08	13.69 ± 6.32	0.5998
Abnormal WBC count	28 (53.85%)	75 (63.56%)	0.232
Platelet count (×10^6^)	145.73 ± 133.06	154.56 ± 116.85	0.6643
Abnormal CRP	50 (96.15%)	114 (96.61%)	0.882
Serum lactate > 2 mmol/L	36 (69.23%)	94 (79.66%)	0.140
Bloodstream infection	27 (51.92%)	75 (63.56%)	0.154
Intravascular device	45 (86.54%)	115 (97.46%)	0.005
Continuous renal replacement therapy (CRRT)	14 (26.92%)	27 (22.88%)	0.570
Extracorporeal membrane oxygenation (ECMO)	0 (0%)	5 (4.24%)	0.132
Mechanical ventilation	40 (76.92%)	116 (98.31%)	0.0001
Septic shock	43 (82.69%)	114 (96.61%)	0.002
Outcomes and therapy			
Steroid therapy	37 (71.15%)	117 (99.15%)	0.0001
Tocilizumab	0 (0%)	39 (33.05%)	0.0001
Transfer to ICU	1.69 ± 4.64	2.58 ± 4.72	0.2594
Length of ICU-stay	28.33 ± 19.63	21.2 ± 11.98	0.0042
Length of hospitalization	29.35 ± 19.71	23.81 ± 15.67	0.0522
Discharge status			0.005
Discharged	9 (17.31%)	24 (20.34%)	
Transferred for other hospitals	6 (11.54%)	1 (0.85%)	
Died	37 (71.15%)	93 (78.81%)	
Mortality at 30 days	26 (50.00%)	75 (63.56)	0.097
Mortality more than 30 days (overall inpatient mortality)	37 (71.15%)	93 (78.81%)	0.489

SOFA: Sequential organ failure assessment, WBCs: white blood cells, CRP: C-reactive protein, and ICU: intensive care unit.

**Table 2 healthcare-11-00487-t002:** COX hazard regression analysis of factors associated with overall mortality.

Variables	Hazard Ratio	CI 95%	*p* Value
COVID-19 infection	1.79	1.02–3.15	0.043
Age	1.01	0.1–1.02	0.119
SOFA at positive culture time	1.12	1.07–1.16	0.000
Steroid therapy	0.5	0.21–1.2	0.119
Continuous renal replacement therapy (CRRT)	0.72	0.46–1.13	0.158
Chronic obstructive pulmonary disease (COPD)	0.57	0.28–1.19	0.137
Hypertension	0.61	0.4–0.94	0.027
Abnormal WBC count	1.57	1.04–2.37	0.033
Chronic kidney disease (CKD)	2.01	1.26–3.19	0.003

CI: confidence interval, SOFA: sequential organ failure assessment, and WBCs: white blood cells.

**Table 3 healthcare-11-00487-t003:** The clinical characteristics of patients with a documented MDR-AB infection with or without a bloodstream infection.

Variables	No Bloodstream Infection *n* = 66	Bloodstream Infection *n* = 94	*p* Value
Age	63.79 ± 16.51	62.04 ± 18.98	0.5351
Gender			0.895
Female (F)	24 (35.29%)	35 (34.31%)	
Male (M)	44 (64.71%)	67 (65.69%)	
Body mass index (*n* = 160)	28.52 ± 6.77	29.3 ± 7.65	0.5116
Body mass index category			0.560
Underweight or normal	17 (25.76%)	20 (21.28%)	
Overweight	29 (43.94%)	38 (40.43%)	
Obese	20 (30.30%)	36 (38.30%)	
Infected with COVID-19	43 (63.24%)	75 (73.53%)	0.154
Pregnancy	0 (0%)	5 (4.90%)	0.064
Comorbidities			
More than two comorbidities	24 (35.29%)	22 (21.57%)	0.048
Heart failure	5 (7.35%)	7 (6.86%)	0.903
Hypertension	51 (75.00%)	61 (59.80%)	0.041
Diabetes Mellitus (DM)	52 (76.47%)	64 (62.75%)	0.060
Chronic kidney disease (CKD)	17 (25.00%)	20 (19.61%)	0.404
Chronic liver disease (CLD)	3 (4.41%)	4 (3.92%)	0.875
Asthma	4 (5.88%)	4 (3.92%)	0.554
Chronic obstructive pulmonary disease (COPD)	10 (14.71%)	7 (6.86%)	0.095
Malignancy	2 (2.94%)	5 (4.90%)	0.528
Clinical features			
SOFA at time of admission	4.91 ± 4.27	5.21 ± 3.96	0.6462
SOFA at positive culture time	12.07 ± 6.40	15.07 ± 6.40	0.0032
Abnormal WBCs count	44 (64.71%)	59 (57.84%)	0.370
Platelet count (×10^6^)	156.47 ± 113.44	148.78 ± 127.37	0.6874
Abnormal CRP	65 (95.59%)	99 (97.06%)	0.611
Serum lactate > 2 mmol/L	47 (69.12%)	83 (81.37%)	0.065
Intravascular device	61 (89.71%)	99 (97.06%)	0.046
Continuous renal replacement therapy (CRRT)	18 (26.47%)	23 (22.55%)	0.558
Extracorporeal membrane oxygenation (ECMO)	0 (0%)	5 (4.90%)	0.064
Mechanical ventilation	59 (86.76%)	97 (95.10%)	0.053
Septic shock	60 (88.24%)	97 (95.10%)	0.099
Outcomes and therapy			
Steroid therapy	58 (85.29%)	96 (94.12%)	0.054
Tocilizumab	13 (19.12%)	26 (25.49%)	0.333
Transfer to ICU	2.25 ± 4.11	2.34 ± 5.07	0.8996
Length of ICU-stay	22.56 ± 13.47	23.95 ± 16.09	0.5575
Length of hospitalization	24.84 ± 13.91	25.95 ± 19.04	0.6796
Discharge status			0.307
Discharged	16 (23.53%)	17 (16.67%)	
Died	48 (70.59%)	82 (80.39%)	
Transferred for other hospitals	4 (5.88%)	3 (2.94%)	
Mortality at 30 days	39 (57.35%)	62 (60.78%)	0.655
Mortality more than 30 days (overall mortality)	48 (70.59%)	82 (80.39%)	0.140

SOFA: Sequential organ failure assessment, WBCs: white blood cells, CRP: C-reactive protein, and ICU: intensive care unit.

**Table 4 healthcare-11-00487-t004:** Multivariate analysis of risk factors associated with acquisition of a bloodstream infection.

Variables	Odds Ratio	CI 95%	*p* Value
COVID-19 infection	2.06	0.9–4.72	0.087
Diabetes Mellitus (DM)	0.61	0.26–1.39	0.236
Abnormal WBCs count	0.55	0.25–1.2	0.131
Abnormal CRP	5.31	0.46–61.62	0.182
Intravascular device	3.05	0.54–17.21	0.207
SOFA at positive culture time	1.13	1.06–1.21	0.001
Continuous renal replacement therapy (CRRT)	0.52	0.22–1.19	0.119

CI: Confidence interval, WBCs: white blood cells, CRP: C-reactive protein, and SOFA: sequential organ failure assessment.

## Data Availability

Data are available based upon request from KFSH.

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
