# Peer review of "The Impact of Multidrug-Resistant *Acinetobacter baumannii* Infection in Critically Ill Patients with or without COVID-19 Infection"

_healthcare, 2023, doi:10.3390/healthcare11040487_

Round 1
Reviewer 1 Report
COVID-19 and antimicrobial resistance (AMR) are both urgent threats that have challenged the health care system worldwide. The authors carried out a retrospective cohort study to evaluate the clinical outcomes of multi-drug resistant Acinetobacter baumannii infections (MDR- AB) in patients admitted to intensive care unit (ICU) with or without COVID-19 infection. They also identified the risk factors for mortality and blood stream infection in the study population, respectively. The study provides real world data regarding the relationship between CVOID-19 and AMR in a large hospital in Saudi Arabia and has a great interest for clinical practice. The study design and methods are overall appropriate. However, there are some concerns that I hope the authors can address:
1. Although overall the writing is fine, the manuscript needs some help with English language and grammar. Some sentences are difficult to read and understand.
2. In the last paragraph of Introduction section, the authors stated that one of their aims was to “determine the risk factors associated with acquiring multidrug-resistant Acinetobacter baumannii (MDR-AB)”. First, given that all the 170 patients in the study were patients with MDR-AB and there were not any control patients without MDR-AB, it is impossible to determine the risk factors associated with MDR-AB development in this study. Secondly, there is not any data presented in the manuscript corresponding to this aim. Therefore, this aim should be removed from the manuscript.
3. One of the study inclusion criteria was that patients have MDR-AB infection. Could the authors provide some information about those MDR-AB infections, such as their infection sites? How many patients had MDR-AB blood stream infections?
4. Of the 170 patients in the study, 118 were admitted to the ICU due to COVID-19 and the remaining 52 patients were admitted for other reasons. What were other reasons? Could the authors provide the reasons for those 52 patients being admitted to ICU? This information could be helpful for readers to better understand this group of patients.
5. Two multivariable analyses were performed in the study, including one Cox regression analysis on mortality and one logistic regression analysis on blood stream infection. It is not clear whether the authors started each multivariable analysis with univariate analysis first. Please clarify it. If the authors did, could the authors provide a bit detail on the variable selection from univariable analysis to multivariable analysis? For instance, what p-value on univariate analysis was chosen to select variables to construct the initial multivariable model for each outcome?
6. Abnormal WBC counts and abnormal CRP are two important laboratory variables in the data analyses, but their definitions are missing in the manuscript. Please provide them. What are cut-off values used for them?
7. In Table 1, for patients with overall in-patient mortality, could the authors provide the days from their ICU admission to their death?
8. Covid-19 infection was included in the multivariate analysis of risk factors associated with developing blood stream infection (Table 4). In my opinion, it may not be appropriate to do so as the study does not provide the information regarding the timing of acquiring COVID-19 and acquiring MDR-AB infections. MDR-AB infection could be MDR-AB blood stream infection and if COVID-19 infection occurred after MDR-AB blood stream infection, even just for some patients, then it is not appropriate to look at COVID-19 as a potential risk factor for acquiring blood stream infection. If this is the case, this factor should be removed from the multivariate analysis to identify the risk factors for blood stream infection.
9. In Discussion, the authors mainly focused on comparing the study findings with those from other studies, could the authors provide some deeper insights into the patients with MDR-AB and COVID-19 such as the management of this superinfection?
10. In Figure 1, I would suggest changing the Axis title from “Analysis time” to “Days after ICU admission”, and changing the “COVIDSTATUS=0” to “COVID STATUS= No” and “COVIDSTATUS=1” to “CVOID STATUS= Yes”.
11. There is a description about mortality rate in the text as the following: “Overall inpatient mortality was not statistically different by COVID-19 status; 21.19% for the COVID-19 group and 28.85% for non-COVID-19 group with a p-value = 0.489.” The 21.19% and 28.85% mentioned were actually survival rates, not mortality rates. To avoid a confusion, please either change them to mortality rates or indicate that they are survival rates.
12. Lastly, I would suggest adding page numbers and line numbers in the manuscript.
Reviewer 2 Report
The manuscript presented by Alenazi et al. describes the impact of MDR A. baumannii infection in critically ill patients in King Fahad Specialist Hospital in Tabuk, Saudi Arabia. The results seem interesting and may be useful to the wider scientific community dealing with subject of MDR infections cohort studies in the light of COVID-19 infection. Although the results may be useful, the manuscript contains several shortfalls.
Chapter 1. Introduction, first and second sentences: Authors should carefully revise this first sentences. A. baumannii is not an infection, it is bacterium, opportunistic pathogen, causative agent of infections. It is not correct to designate pathogen as an infection, please correct this. English should be carefully revised (by a native speaker or a professional). Generally whole chapter introduction seems like that authors did not expressed their thoughts in a best way, but rather it looks like copy-paste form cited literature without consistency. It is not easy to follow the text.
Chapter 2.1. Study design and patient selection: I suggest to the authors to try to present this part in more transparent way (e.g. divide into parts determined information), and not just write down all at once. The sentence: “The following information was determined” seems endless.
Table 1. Please explain why for variable Body mass index n=160, and not n=170? The total number of patients included in the study is 170.
Figure 1. What is the measure unit for Analysis time on x-axis, days? And what is presented on y-axis? Cumulative survival? Both axes should be properly designated, and followed with adequate explanation in the text.
Chapter References should be carefully revised. References 2 and 3 are one reference, the same applies to references 4, 5 and 6; 85, 86 and 87; etc. The last cited reference in the text is 40, and there are 89 references listed, please correct this.
